# Quasiparticle Andreev scattering in the $\nu = 1/3$ fractional quantum Hall regime

P. Glidic [1,3], O. Maillet [1,3], C. Piquard [1], A. Aassime[1], A. Cavanna [1], Y. Jin[1], U. Gennser [1], A. Anthore [1,2] ✉ & F. Pierre [1] ✉

The scattering of exotic quasiparticles may follow different rules than electrons. In the fractional quantum Hall regime, a quantum point contact (QPC) provides a source of quasiparticles with field effect selectable charges and statistics, which can be scattered on an 'analyzer' QPC to investigate these rules. Remarkably, for incident quasiparticles dissimilar to those naturally transmitted across the analyzer, electrical conduction conserves neither the nature nor the number of the quasiparticles. In contrast with standard elastic scattering, theory predicts the emergence of a mechanism akin to the Andreev reflection at a normal-superconductor interface. Here, we observe the predicted Andreev-like reflection of an $e/3$ quasiparticle into a $-2e/3$ hole accompanied by the transmission of an $e$ quasielectron. Combining shot noise and cross-correlation measurements, we independently determine the charge of the different particles and ascertain the coincidence of quasielectron and fractional hole. The present work advances our understanding on the unconventional behavior of fractional quasiparticles, with implications toward the generation of novel quasi-particles/holes and non-local entanglements.

How do exotic quasiparticles modify when one tries to manipulate them? A conventional free electron incident upon a local barrier can be either elastically transmitted or reflected with different probability amplitudes, matching a beam splitter behavior with electron quantum optics applications[1]. However, this simple picture may be drastically altered with unconventional quasiparticles, such as the emblematic anyons in the fractional quantum Hall (FQH) regime[2]. Fractional quasiparticles could undergo markedly different transmission mechanisms across a barrier, where the number and even the nature of the quasiparticles may change. Such behaviors emerge when the barrier is set to favor the transmission of a type of particles that is different from the incident ones. This is specifically expected in the fractional quantum Hall regime at filling factor $\nu = (2n+1)^{-1}$ ($n \in \mathbb{N}$), when individual quasiparticles of charge $\nu e$ are impinging on an opaque barrier transmitting quasielectrons of charge $e$. In a dilute beam, where no multiple quasiparticles are readily available for bunching into a quasielectron, theory predicts that the missing $(1-\nu)e$ can be supplied in an Andreev-

like scenario involving the correlated reflection of a $-(1-\nu)e$ quasihole[3]. This can also be seen as the quasiparticle transmission coinciding with the excitation of $(1/\nu - 1)\nu e$ quasiparticle-quasihole pairs split between the two outputs (see Fig. 1a for an illustration at $\nu = 1/3$).

A versatile investigation platform is realized by two quantum point contacts (QPC) in series along a fractional quantum Hall edge channel, combined with noise characterizations[4–9]. The first QPC here implements a source of dilute quasiparticles, impinging one at a time on the second 'analyzer' QPC playing the role of the barrier.

We presently investigate at $\nu = 1/3$ such Andreev-like behavior schematically illustrated in Fig. 1a,b. Fractional quasiparticles of charge $e/3$ are separately emitted at the upstream source QPC, which is set to this aim in the so-called weak back-scattering (WBS) regime[3,10,11] and submitted to a voltage bias. After propagating along a short chiral edge path, the quasiparticles individually arrive at the analyzer QPC set in the opposite strong back-scattering (SBS) regime that favors the

[1]Université Paris-Saclay, CNRS, Centre de Nanosciences et de Nanotechnologies, 91120 Palaiseau, France. [2]Université Paris Cité, CNRS, Centre de Nanosciences et de Nanotechnologies, F-91120 Palaiseau, France. [3]These authors contributed equally: P. Glidic, O. Maillet. ✉e-mail: anne.anthore@c2n.upsaclay.fr; frederic.pierre@cnrs.fr

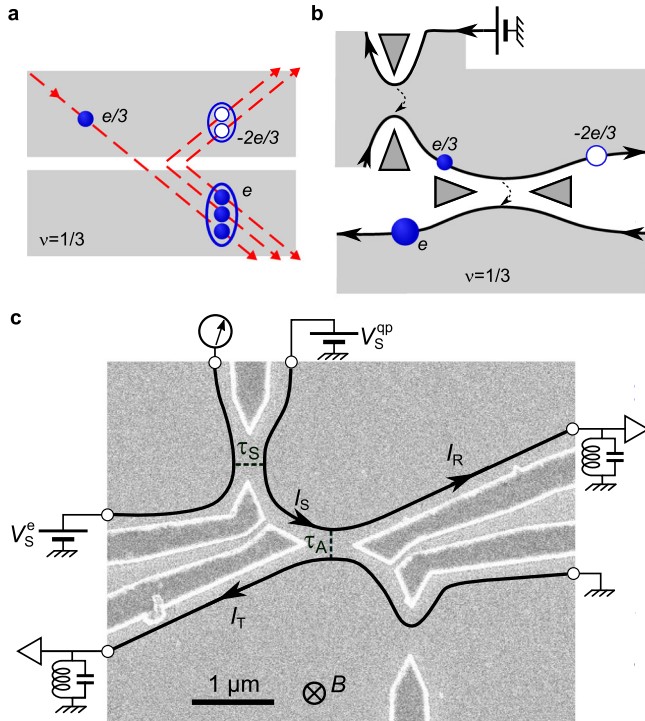

**Fig. 1 | Quasiparticle Andreev reflection in a source-analyzer setup at $\nu = 1/3$.**
**a** Andreev mechanism. An incident $e/3$ quasiparticle is transmitted as an $e$ quasielectron and Andreev reflected as a $-2e/3$ quasihole. The Andreev process can be pictured as the excitation of two $e/3$ quasiparticle-quasihole pairs and the incident quasiparticle bunching together. **b** Setup schematic in Andreev reflection configuration. The top-left `source' QPC is set in the weak back-scattering (WBS) regime and voltage biased from the top to emit $e/3$ quasiparticles toward the central `analyzer' QPC. The latter is tuned in the strong back-scattering (SBS) regime favoring the transmission of quasielectrons. **c**, Electron micrograph of the measured Ga(Al)As device. The current propagates along chiral edge channels shown as black lines. The gate defined QPCs are tuned by field effect. The source is biased with $V_S^{qp}$ at $V_S^e = 0$ and $1 - \tau_S \ll 1$ ($V_S^e$ at $V_S^{qp} = 0$ and $\tau_S \ll 1$) to produce a dilute current of quasiparticles $I_S = (1 - \tau_S)V_S^{qp}/(3h/e^2)$ (of quasielectrons $I_S = \tau_S V_S^e/(3h/e^2)$). Setting $V_S^{qp} = V_S^e$ allows for a direct voltage bias of the analyzer.

transmission of quasielectrons[3,10,11]. Whereas for a directly voltage biased QPC in the SBS regime, a quasielectron can be formed from the bunching of three available $e/3$ quasiparticles, we are here in the presence of a single incident quasiparticle that carries only a third of the required electron charge. In principle, individual $e/3$ quasiparticle tunneling could emerge as the dominant process. However, as presently observed, a different scenario akin to Andreev reflection is expected, where the missing $2e/3$ charge is sucked in from the incident edge channel to form the transmitted quasielectron. The incident fractional quasiparticle is effectively converted into a quasielectron and a $-2e/3$ fractional hole.

This mechanism was coined 'Andreev'[3], by analogy with the standard Andreev reflection of an electron into a hole to transmit a Cooper pair across a normal metal-superconductor interface[12]. Note however that the QPC is not here at an interface with a superconductor, nor with a different fractional quantum Hall state (see Refs. [13–15] for another, different kind of Andreev-like reflection at such interfaces submitted to a voltage bias). Furthermore, whereas in a standard Andreev reflection electron and hole excitations have the same energy, here energy conservation imposes that the energy of the incident quasiparticle redistributes between transmitted quasielectron and reflected quasihole. The energy of the reflected quasihole is thus lower than that of the incident quasiparticle. Finally, we point out that the present Andreev-like mechanism takes place in a

fully spin-polarized electronic fluid, in contrast with a standard Andreev reflection where two electrons of opposite spins are combined to form a spin-singlet Cooper pair.

Experimentally, an earlier source-analyzer investigation appeared to contradict this scenario[16]. Indeed, the transmitted charge was there found to approach $e/3$ across the opaque barrier, identical to the charge of the incident quasiparticles, instead of $e$ for Andreev processes (see Ref. [17] for a follow-up paper that mitigates this conclusion, by the observation of an increase in the transmitted charge as the temperature is reduced). Possibly, the $e/3$ quasiparticles have been altered during the very long propagation distance of ∼100 $\mu m$ between source and analyzer QPCs. Here, with a short 1.5 $\mu m$ path (see Fig. 1c), we recover the predicted transmitted charge $e$, three times larger than the simultaneously determined charge of the incident quasiparticles. Moreover, we directly observe the Andreev correlations between transmitted quasielectron and reflected $-2e/3$ fractional hole, through the revealing measurement of the current cross-correlations between the two outputs of the analyzer QPC.

## Results

### Device and setup

The measured sample is shown in Fig. 1c (see Methods for large-scale pictures). It is patterned on a high-mobility Ga(Al)As two-dimensional electron gas (2DEG) of density $1.2 \times 10^{11}$ cm$^{-2}$. The device is cooled at a temperature $T \approx 35$ mK (see Methods for supplementary data at $T \approx 15$ and $60$ mK), and immersed in a perpendicular magnetic field $B \approx 13.5$ T near the center of a 2 T wide quantum Hall resistance plateau $R_H = 3h/e^2$ ($\nu = 1/3$). In this FQH regime, the electrical current propagates along each edge in a single chiral channel, as schematically depicted by black lines with arrows in Fig. 1b,c (see Methods for tests of this picture). These edge channels are measured and biased through large ohmic contacts of negligible resistance located 150 $\mu m$ away from the central part (symbolized as open black circles, see Methods for the actual shape). The heart of the device is composed of two active QPCs (out of three nanofabricated ones) separately tuned by field effect with the voltages applied to the corresponding aluminum split gates deposited at the surface (darker areas with bright edges). The top-left QPC (or, alternatively, the bottom-right QPC) plays the role of the quasiparticle source, whereas the central QPC is the downstream analyzer. The auto- and cross-correlations of the currents $I_T$ and $I_R$ emitted from the analyzer QPC are capital for the separate tunneling charge characterization across source and analyzer, as well as for providing direct signatures of Andreev processes. They are measured using homemade cryogenic amplifiers[18,19], in a 40 kHz bandwidth centered on the resonant frequency 0.86 MHz of essentially identical tank circuits along the two amplification chains.

### Quantum point contact characterization

We first determine the characteristic tunneling charges across the source and analyzer through standard shot noise measurements[20–22], involving a direct voltage bias of the considered QPC (as opposed to a dilute beam of quasiparticles, see below). For the analyzer, such characterization must therefore be performed in a specific measurement, distinct from the observation of Andreev processes. This is achieved without changing any gate voltages susceptible to impact the analyzer's tuning, by using the same bias voltage for the two input channels of the source QPC ($V_S^{qp} = V_S^e$, see Fig. 1c). In the present work, the analyzer QPC is set in the SBS regime, i.e. with a low transmission ratio $\tau_A \equiv I_T/I_S$ for which theory predicts the transmission of quasielectrons[10,11]. Accordingly, we focus here on tunings of the analyzer displaying this canonical behavior, such as shown in Fig. 2b. The filled (open) blue circles display the measured excess transmitted (reflected) noise $\langle \delta I_{T(R)}^2 \rangle_{exc} \equiv \langle \delta I_{T(R)}^2 \rangle(V_S^{qp} = V_S^e) - \langle \delta I_{T(R)}^2 \rangle(0)$ versus bias

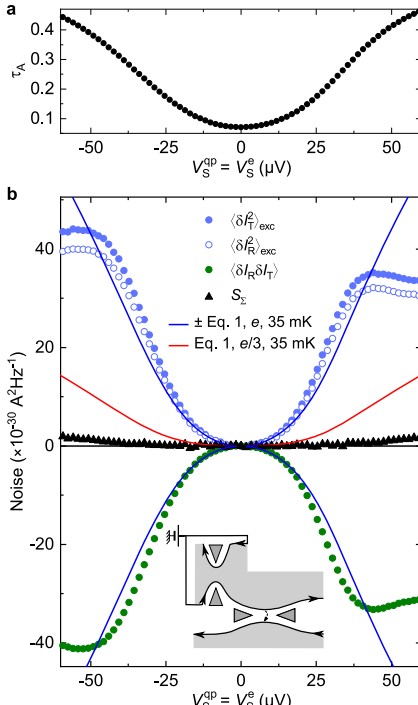

**Fig. 2 | Characterization of analyzer QPC, from transmission (a) and noise (b) vs direct voltage bias $V_S^{qp} = V_S^e$. a** Transmission ratio $\tau_A \equiv I_T/I_S$. **b** Measurements of the auto- and cross-correlations of the transmitted ($I_T$) and reflected ($I_R$) currents are shown as symbols. For small enough $\tau_A \lesssim 0.3$ ($|V_S^{qp}| < 35\,\mu V$, see (**a**)), the different noise signals corroborate the expected tunneling charge $e$ (blue lines) in marked difference with $e/3$ predictions (red line). At higher $\tau_A \gtrsim 0.3$, the relatively smaller noise is consistent with the onset of the expected transition toward $e/3$. The noise sum $S_\Sigma \equiv \langle \delta I_T^2 \rangle_{exc} + \langle \delta I_R^2 \rangle_{exc} + 2\langle \delta I_T \delta I_R \rangle$, corresponding to the excess shot noise across the presently unbiased source, remains essentially null.

voltage. The tunneling charge $e^*$ is determined by comparing with the standard shot noise expression[22,23]:

$$S_{sn} = 2e^* \frac{\tau(1-\tau)V}{R_H}\left[ \coth\frac{e^*V}{2k_B T} - \frac{2k_B T}{e^*V} \right],\qquad(1)$$

with $\tau$ the ratio of transmitted over incident dc currents. The positive blue and red continuous lines display the predictions of Eq. (1) for $e^* = e$ and $e/3$, respectively, at $T = 35$ mK and using the simultaneously measured $\tau_A$ shown in Fig. 2a. The negative blue line shows $-S_{sn}$ for $e^* = e$. Note that $\tau_A$ strongly increases with the applied bias voltage, which also usually drives a transition from $e^* = e$ (at $\tau_A \ll 1$) to $e/3$ (at $1 - \tau_A \ll 1$)[10,11,24]. Correspondingly, an agreement is here found with $e^* = e$ only at low enough bias voltages ($|V| < 35\,\mu V$), for which $\tau_A$ is not too large ($\tau_A \lesssim 0.3$). An important experimental check consists in confronting $\langle \delta I_T^2 \rangle_{exc}$ with both the reflected excess noise $\langle \delta I_R^2 \rangle_{exc}$ and the possibly more robust cross-correlation signal[25] $\langle \delta I_T \delta I_R \rangle$. We find that the three measurements match each other, within the experimental gain calibration accuracy (Methods), thereby corroborating the extracted value of $e^*$. Equivalently, the sum $S_\Sigma \equiv \langle \delta I_T^2 \rangle_{exc} + \langle \delta I_R^2 \rangle_{exc} + 2\langle \delta I_T \delta I_R \rangle$ (black symbols) mostly does not depend on bias voltage, as expected in the absence of shot noise across the upstream source QPC. Indeed, local charge conservation and the chirality of electrical current directly imply the identity $S_\Sigma = \langle \delta I_S^2 \rangle_{exc}$, independently of the downstream analyzer (the weak positive $S_\Sigma$ that can be seen at large bias is in fact a small noise induced at the source, see Methods). In the source-analyzer configuration, this identity will be essential for the characterization of the tunneling charge across the source QPC simultaneously with the measurement of the main cross-

correlation signal, by confronting $S_\Sigma$ with Eq. (1) (see Fig. 3a,b, and also Methods)[26].

## Observation of Andreev-like reflection of fractional quasiparticles

The source is now activated with the device set in the regime where Andreev reflections are predicted, and direct signatures of this process are observed. For this purpose, the source QPC is tuned in the WBS regime and biased through $V_S^{qp}$ ($V_S^e = 0$). As shown in Fig. 3a, the backscattering probability $1 - \tau_S = I_S R_H/V_S^{qp}$ (inset) remains very small (<0.05), and $S_\Sigma$ (symbols in main panel) matches the prediction of Eq. (1) for $e^* = e/3$ using $T = 35$ mK (red line). As a side note, we point out the decrease of $\tau_S$ with $I_S$ and thus the applied bias, which although not expected theoretically[11,27] is frequently observed experimentally at high transmission (see Methods and e.g. ref. [28]).

With the upstream generation of a highly dilute beam of $e/3$ quasiparticles established, we turn to the characterization of the transport mechanism across the downstream analyzer kept in the SBS regime previously characterized. The blue symbols in Fig. 3c display the measured excess shot noise on the current transmitted across the analyzer $\langle \delta I_T^2 \rangle_{exc}$, over a range of $I_T$ corresponding to that of $I_S$ in panel (a) ($I_T = \tau_A I_S$, see inset for $\tau_A$). The shot noise data closely follow the slope of $2e|I_T|$ (dashed blue line at $|I_T| > 1$ pA) denoting the Poissonian transfer of $1e$ charges, different from the $e/3$ charge of the incident quasiparticles. This corresponds to Andreev processes, in marked contrast with the slope of $2(e/3)|I_T|$ (dashed red line) approached in the dilute beam limit in the pioneer experiment ref. [16], and consistent with the different trend described in the follow-up article ref. [17]. Note that the small thermal rounding, at $|I_T| < 1$ pA, matches the displayed generalization of Eq. (1) where we used the source bias voltage ($V_S^{qp}$) and tunneling charge ($e/3$) in the $e^*V/k_B T$ ratios (see Eq. (5) in Methods).

As emphasized in ref. [3], a key feature of Andreev processes is that the transmitted and reflected currents are correlated, for which the measurement of $\langle \delta I_R \delta I_T \rangle$ provides an unambiguous signature. Since the Andreev transfer of a charge $e$ is associated with the reflection of a charge $-2e/3$, theory predicts the straightforward connection[3]:

$$\langle \delta I_R \delta I_T \rangle = -(2/3)\langle \delta I_T^2 \rangle_{exc},\qquad(2)$$

where the factor of $-2/3$ directly corresponds to the ratio between tunneling and reflected charges. Accordingly, the slope of $-(2/3)2e|I_T|$ (dashed black line at $|I_T| > 1$ pA) is compared in Fig. 3c with the measurements of $\langle \delta I_R \delta I_T \rangle$ shown as green symbols. The observed quantitative match most directly attests of the underlying Andreev-like mechanism (see Methods for different device tunings and temperatures).

## Noise signal with incident quasielectrons

An instructive counterpoint, clarifying the specificity of the above Andreev signatures, is obtained by tuning the source QPC in the SBS regime with a tunneling charge $e^* = e$. In this configuration, the source is voltage biased by $V_S^e$ (with $V_S^{qp} = 0$). As shown in Fig. 3b, the source shot noise obtained from $S_\Sigma$ follows the prediction of Eq. (1) for $e^* = e$ and $T = 35$ mK (blue line) as long as the transmission remains low enough ($\tau_S \lesssim 0.3$). Noise data points displayed as full (open) symbols in Fig. 3b, d correspond to $\tau_S < 0.3$ ($\tau_S > 0.3$). Whereas $\langle \delta I_T^2 \rangle_{exc} \approx 2e|I_T|$ indicates the same $1e$ tunneling charge across the analyzer as in the previously discussed Andreev regime, it here also trivially corresponds to the charge of the incident particles. In marked contrast to Andreev processes, the cross-correlations $\langle \delta I_R \delta I_T \rangle$ are no longer negative, but relatively small and positive. The continuous blue and green lines in (d) display the predictions for non-interacting electrons at $T = 35$ mK (see Eqs. (6) and (7) in Methods). While no signal would be expected in the Poisson limit, note the prediction of appreciable negative cross-correlations (green line). This results from the rapidly growing $\tau_S$ (see

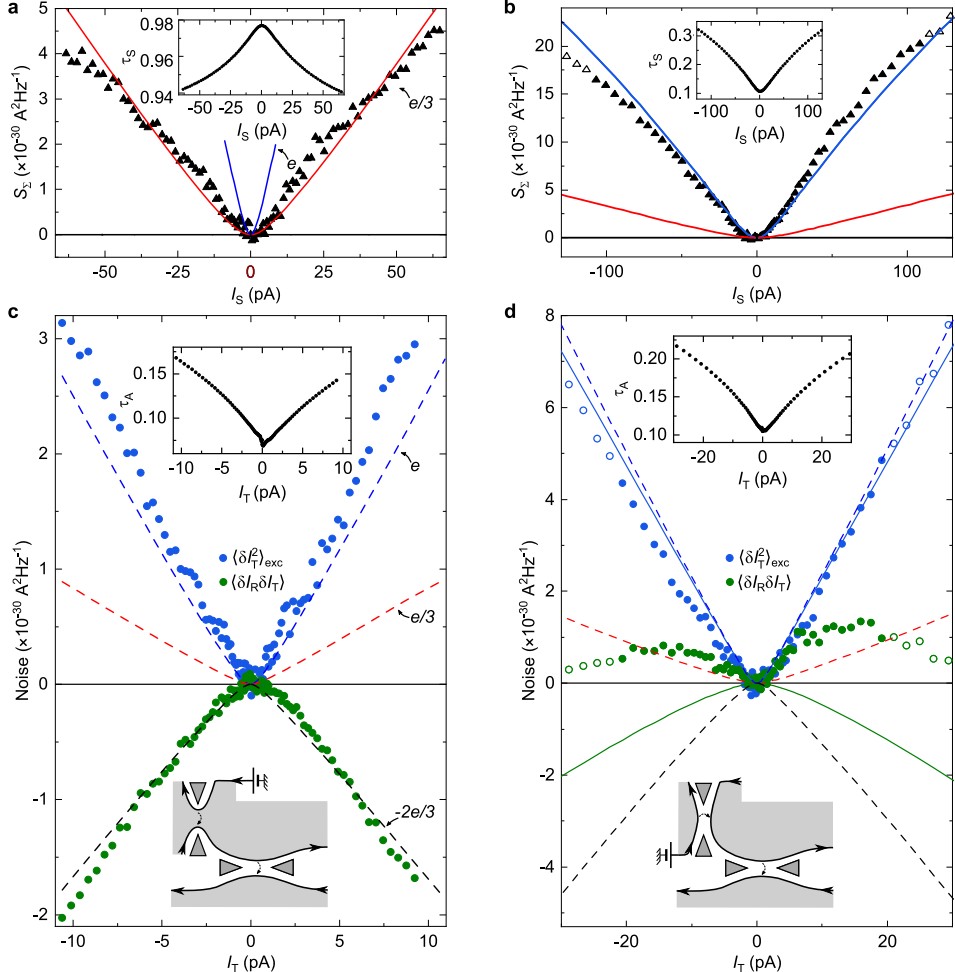

**Fig. 3 | Noise observation of Andreev reflection. a, b** Simultaneous characterization of the source set in the WBS (**a**) or SBS (**b**) regime (see illustrative bottom inset in (**c**) or (**d**), respectively). Continuous blue and red lines represent the shot noise predictions of Eq. (1) for tunnelings of charge $e$ and $e/3$, respectively, using the measured transmission ratio $\tau_S$ across the source QPC (inset) and $T = 35$ mK. Symbols display measurements of $S_\Sigma$, corresponding to the shot noise across the source. **c, d** Transport mechanism across the analyzer with incident fractional quasiparticles (**c**, using the WBS source shown in (**a**)) or incident quasielectrons (**d**, using the SBS source shown in (**b**)). The simultaneous measurements of $\tau_A \lesssim 0.2$ are shown in the respective top insets (note the higher noise at low $I_T$ due to the

reduced signal). Blue and green symbols in the main panels show, respectively, the excess auto-correlations of the transmitted current and the cross-correlations between transmitted and reflected currents. Open symbols in panels (**b**) and (**d**) correspond to data with $\tau_S \geq 0.3$, for which the source notably deviates from the SBS regime. Dashed blue, red and black lines represent, respectively, a $1e$ shot noise, a $e/3$ shot noise and $-(2/3)$ times the $1e$ shot noise all in the dilute incident beam limit. Continuous lines in (**d**) display the noninteracting electrons' predictions valid at any $\tau_{A,S}$ for $\langle \delta I_T^2 \rangle$ (blue) and $\langle \delta I_R \delta I_T \rangle$ (green), calculated using the measured $\tau_{A,S}$ (see Eqs. (6) and (7) in Methods).

inset of Fig. 3b), which makes it difficult to remain well within the dilute incident beam regime. Whereas the observed positive cross-correlations are not accounted for, suggesting that the role of interactions cannot be ignored (see ref. [29] for positive cross-correlations predicted in the different case of multiple copropagating channels), the contrast with the Andreev signal given by Eq. (2) (dashed black line) is even more striking.

**Additivity of Andreev cross-correlations from opposite sources**
Recently, it was predicted and observed that negative cross-correlations can also develop with dilute incident quasiparticles when both source and analyzer QPCs are set in the same WBS limit[8,26]. This results from the non-trivial braid (double exchange) phase of $2\pi/3$ between $e/3$ quasiparticles[8,30,31], in contrast with the braid phase between quasielectrons and $e/3$ anyons, which has the trivial value $2\pi$, and thus plays no role in Andreev processes (with the analyzer QPC in the SBS limit)[32,33]. We will now show that, beside the fact that they take place in different regimes, exchange-driven and Andreev-like mechanisms can be qualitatively distinguished by using a second

source QPC feeding the same analyzer from the opposite side (bottom-right QPC in Fig. 1c, see schematics in Fig. 4). Indeed, in the exchange-driven tunneling mechanism, each incident quasiparticle leaves behind a trace that affects the tunneling current contribution of the following ones, including in the limit of highly dilute incident beams[2,8,30,31,34]. Specifically, quasiparticles from opposite sources are associated with anyons braiding processes of opposite winding directions that cancel each other (if within a small enough time window) in the relevant total braid phase[30,31]. This results in a dependence of the exchange-driven mechanism on the symmetry between sources. In the language of refs. [8,26], the normalized cross-correlation slope ('P') is reduced by a factor of $\simeq 1.5$ with two symmetric sources. In contrast, the successive Andreev tunnelings are predicted to be independent in the limit of highly diluted incident beams[3]. Consequently, the cross-correlation contributions from the two sources on opposite sides should here simply add up. This distinctive property is demonstrated in Fig. 4. The black symbols display the cross-correlations measured in the presence of two nearly symmetrical diluted beams of $e/3$ quasiparticles impinging on the central analyzer QPC set in the SBS regime.

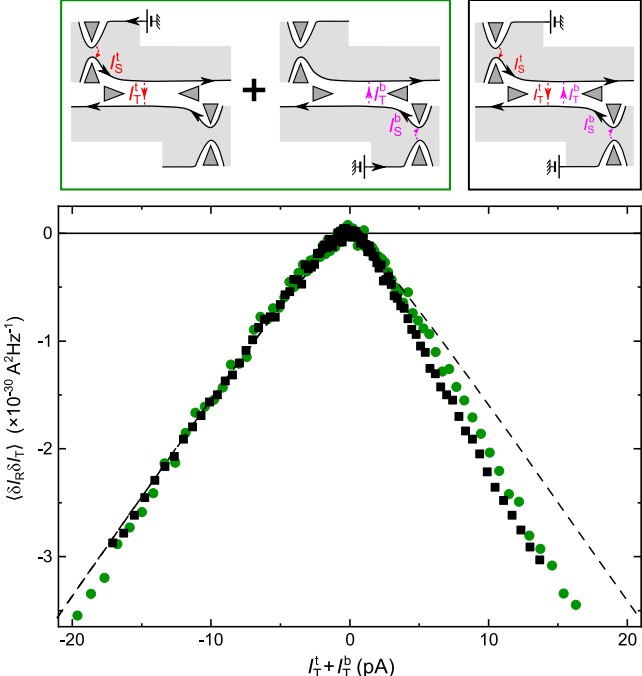

**Fig. 4 | Additivity of Andreev cross-correlations from opposite sources.** The black squares represent the cross-correlations measured with two similar beams of $e/3$ quasiparticles impinging from opposite sides on the central QPC set in the SBS regime (see top-left schematic). The green circles display the sum of the cross-correlations measured sequentially, using separately the top-right or bottom-right QPC as a single source (see top-right schematic). The dashed line shows the predicted cross-correlations for Andreev scatterings, independent of the symmetry between opposite sources. This contrasts with another, symmetry-dependent mechanism based on the unconventional anyon exchange phase occurring with both source and analyzer in the WBS regime[8,26]. See Supplementary Information for a comparison with another gate voltage tuning of the device that exhibits a more canonical behavior at positive tunnel current.

The data is plotted as a function of the sum of the dc tunneling currents originating from the top-left ($I_T^t$) and bottom-right ($I_T^b$) source QPCs, separately determined by lock-in techniques. For a first comparison, the same Andreev prediction previously shown in Fig. 3c is displayed as a dashed line, and found in identically good agreement with the measurement in the presence of two sources. For a most straightforward demonstration, the green symbols display the sum of the two separately measured cross-correlation signals when using solely for the source either the top-left QPC or the bottom-right QPC. The matching between green and black symbols directly shows that the contributions of the two sources simply add up, in qualitative difference with predictions[8] and observations[26] for exchange-driven tunneling processes when all the QPCs are set in the WBS regime.

## Discussion

The present work investigates the emergence of markedly different transport mechanisms for fractional quasiparticles. In the observed Andreev-like scattering at $v = 1/3$, one $e/3$ quasiparticle impinging on a QPC in the SBS regime transforms into a correlated pair made of a transmitted quasielectron and a reflected hole of charge $-2e/3$. In stark contrast with the prominent electron beam splitter picture of QPCs, the number and nature of the quasiparticles are not conserved, with notable implications for envisioned anyonic analogues of quantum optics experiments. Remarkably, the complementary fractional charges of the Andreev-reflected holes might be associated with a distinctive exchange statistics[32,33,35], expanding the range of available exotic quasiparticles for scrutiny and manipulations, and their correlation with the transmitted particle could provides a new knob to

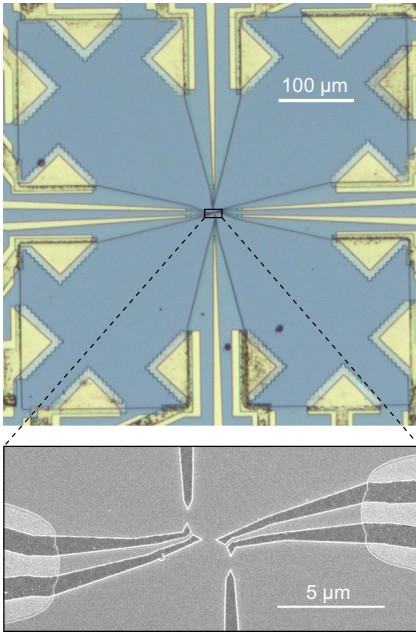

**Fig. 5 | Large-scale sample pictures.** Optical (top) and SEM (bottom) images of the measured device.

generate non-local quantum entanglements. The multiplicity of quasiparticles accessible through the tunings of the fractional filling factor and of the QPCs, suggests that the present observation may generalize into a family of Andreev-like mechanisms, calling for further theoretical and experimental investigations. An illustration at reach is the possible Andreev reflection at $v = 2/5$ of an incident $e/5$ quasiparticle into a hole of charge $-2e/15$ and a transmitted $e/3$ quasiparticle.

## Methods
### Sample
The sample is patterned on a Ga(Al)As heterostructure forming a 2DEG of density $n = 1.2 \times 10^{11}$ cm$^{-2}$ and mobility $1.8 \times 10^6$ cm$^2$ V$^{-1}$s$^{-1}$ at a depth of 140 nm below the surface. Large-scale pictures are shown in Fig. 5. The mesa is defined by wet etching over a depth of about 100 nm (deeper than the Si $\delta$-doping located 65 nm below the surface), using a protection mask made of a ma-N 2403 positive resist patterned by e-beam lithography and etching the unprotected parts in a solution of $H_3PO_4/H_2O_2/H_2O$. The electrical connection to the buried 2DEG is made through large ohmic contacts, realized by the successive deposition of Ni (10 nm) · Au (10 nm) · Ge (90 nm) · Ni (20 nm) · Au (170 nm) · Ni (40 nm) followed by an annealing at 440°C for 50 s in a ArH atmosphere. The lithographic tip to tip distance of the Al split gates used to define the QPCs is 600 nm.

### Experimental setup
The device is operated in a cryofree dilution refrigerator with extensive filtering and thermalization of the electrical lines (see ref. [36] for details). Specific cold $RC$ filters are included near the device, located within the same metallic enclosure screwed onto the mixing chamber: 200 kΩ-100 nF for gate lines, 10 kΩ-47 nF for injection lines, 10 kΩ-1 nF for low-frequency measurement lines.

The lock-in measurements are performed at frequencies below 100 Hz, applying an ac modulation of rms amplitude always below $k_B T/e$. The dc currents $I_S$ and $I_T$ are obtained by integrating with the source bias voltage the corresponding lock-in signal. As an illustrative example, the tunneling current associated with the top-left source ($I_T^t$) is obtained (separately from the tunneling current $I_T^b$ originating from the bottom-right source when the two sources are used

simultaneously) using:

$$I_T^t(V_S^{qp}) = \int_0^{V_S^{qp}} \frac{\partial I_T}{\partial V_S^{qp}} dV_S^{qp}, \qquad (3)$$

where the differential conductance at finite bias voltage is directly given by the lock-in signal measured on port $T$ at the frequency of the ac modulation added to $V_S^{qp}$.

The auto- and cross-correlation noise measurements are performed using two cryogenic amplifiers (see supplementary material of ref. [19] for details) connected to the $T$ and $R$ ports of the device through closely matched $RLC$ tank circuits of essentially identical resonant frequency $\approx 0.86$ MHz (see schematic representation in Fig. 1c). The $RLC$ tanks include home-made superconducting coils of inductance $L_{tk} \approx 250\,\mu$H in parallel with a capacitance $C_{tk} \approx 135$ pF developing along the interconnect coaxial cables, and an effective resistance $R_{tk} \approx 150$ k$\Omega$ (in parallel with $R_H$) essentially resulting from the resistance of the coaxial cables at the lowest temperature stage of the refrigerator. In practice, we integrate the noise signal for 10 s and perform several consecutive voltage bias sweeps (except for temperature calibration), typically between 2 and 12. The displayed noise data is the mean value of these sweeps for the same biasing conditions. Note that the scatter between nearby points adequately indicates the standard error of the displayed mean separately obtained from the ensemble of averaged noise data points (not shown).

## Thermometry

The electronic temperatures at $T > 40$ mK are obtained from a calibrated RuO$_2$ thermometer thermally anchored to the mixing chamber of the dilution refrigerator. In this range, the thermal noise from the sample is found to change linearly with the RuO$_2$ temperature (see also gain calibration of the noise amplification chains). This confirms the good thermalization of electrons in the device with the mixing chamber, as well as the calibration of the RuO$_2$ thermometer. At $T \lesssim 40$ mK, we use the known robust linear dependence of the noise with the electronic temperature to extrapolate from the observed noise slope. The two amplification chains give consistent temperatures, although the difference grows as temperature reduces further away from the calibrated slope, up to 2 mK at the lowest used temperatures $T \approx 15$ mK.

## Noise amplification chains calibration

The gain factors $G_{T,R,TR}^{eff}$, between the power spectral density of current fluctuations of interest and the raw auto/cross-correlations, are obtained from the measurement of the equilibrium noise at different temperatures combined with a determination of the tank circuit parameters.

In a first step, we characterize the tank circuits connected to the device contacts labelled $T$ and $R$. This is achieved through the value of the resonant frequency together with the evolution of the noise bandwidth of the tank in parallel with the known $R_H$ at different filling factors. As a technical note, we mention that correlations between voltage and current noises generated by the cryogenic amplifier can deform the resonance at large $R_H$ and thereby impact the tank parameters extraction. For this purpose, the bandwidth data are taken at sufficiently high temperature ($T \gtrsim 150$ mK) such that these amplifier-induced correlations remain negligible with respect to thermal noise. The obtained tank parameters are summarized in the table within Fig. 6, also showing the fits of the bandwidth vs $R_H$.

In a second step, for our fixed choice of noise integration bandwidth [0.84, 0.88] MHz (which impacts $G_{T,R,TR}^{eff}$), the raw integrated noise is measured at different temperatures $T_{RuO2} > 40$ mK given by a pre-calibrated RuO$_2$ thermometer thermally anchored to the mixing chamber (see Thermometry above). From the fluctuation-dissipation

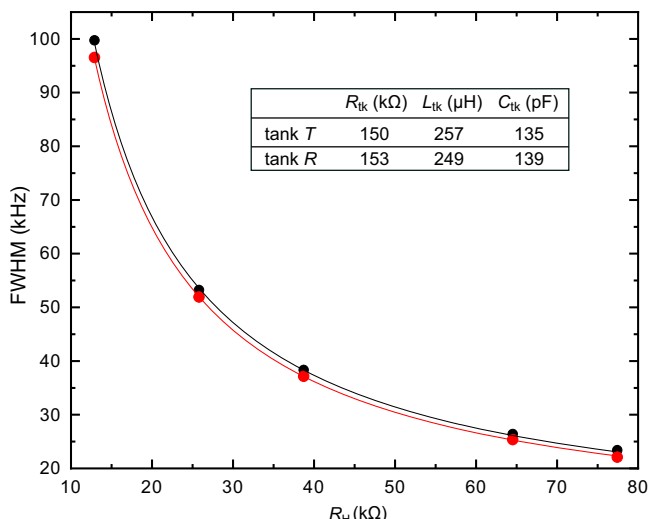

**Fig. 6 | Tank circuits characterization.** Full Width at Half Maximum (FWHM) of the measured tank resonance in the noise signal, as a function of the sample's resistance $R_H$. Black (red) dots represent the FWHM of tank $T$ ($R$) measured with the device set on resistance plateaus of known $R_H$ ($\nu \in \{2, 1, 2/3, 2/5, 1/3\}$). Solid lines show $1/2\pi C_{tk}R$, with $1/R \equiv 1/R_H + 1/R_{tk}$. The values of $R_{tk}$ and $C_{tk}$ used as fit parameters are recapitulated in the table together with the inductances $L_{tk}$ given by the resonant frequencies.

relation, we have:

$$G_{T,R}^{eff} = \frac{s_{T,R}}{4k_B\left(1/R_{tk}^{T,R} + 1/R_H\right)}, \qquad (4)$$

with $s_{T(R)}$ the temperature slope of the raw integrated noise on measurement port $T$ ($R$), and $R_{tk}^{T(R)}$ the effective parallel resistance describing the dissipation in the tank circuit connected to the same port. Note that the only required knowledge of the tank is here $R_{tk}$, whose impact remains relatively small compared to that of $R_H$ even at $\nu = 1/3$. In particular, the relation Eq. (4) does not involve the tank bandwidth nor our choice of frequency range used to integrate the noise signal (although the slopes $s_{T,R}$ depend on these parameters). In contrast, the cross-correlation gain $G_{TR}^{eff}$ can also be reduced by an imperfect matching between the tanks (see e.g. the supplementary material of ref. [26] for a detailed presentation). However, for our tank parameters this reduction is negligible ($< 0.5\%$) and $G_{TR}^{eff} \simeq \sqrt{G_T^{eff} G_R^{eff}}$.

The above main calibration is checked with respect to a thermal calibration at $\nu = 2$ where the relative impact of $R_{tk}$ is reduced. Then, using the simple $RLC$ model of the tank circuits as recapitulated in the table in Fig. 6, the $\nu = 2$ calibration can be converted into $G_{T,R}^{eff}$ at $\nu = 1/3$ for the corresponding (different) integration bandwidth and $R_H$. This control procedure, relying in its first (second) step less (more) heavily on the knowledge of the tank circuits, gives compatible $G_{T,R}^{eff}$ at an accuracy better than 7%: Through this procedure $G_{T(R)}^{eff}$ is found to be 6.8% (2.0%) higher than with the main calibration above (note that this could account for the small difference between the auto-correlations in the transmitted and reflected current in Fig. 2b). In a second cool-down of the same sample, this check calibration at $\nu = 2$ was used to correct for a small ($\lesssim 2\%$) change in the gains of the cryogenic amplifiers.

## Quantum point contacts

Typical sweeps of the transmission ratio at zero dc bias voltage as well as the differential fraction of the transmitted current in the presence of a dc bias of $\approx 40\,\mu$V are shown in Fig. 7 versus gate voltage for the two sources and the analyzer QPCs. The down and up

arrows points to the regions used for tuning the QPCs in, respectively, the SBS and WBS regime. Note that the actual tuning of each QPC is also impacted by the choice of voltages of the other nearby gates. Note also that whereas both gates are simultaneously swept for the analyzer, only the upper (lower) gate is swept for the source top-left (bottom-right) QPC. This reduces the impact on the central analyzer QPC of changing the source's tuning from SBS to WBS.

Intriguingly, the central analyzer QPC requires more negative gate voltages to be fully closed than the two rather similar source QPCs. This different behaviour, systematically observed on several devices of the same chip, may be due to the different orientation of the analyzer QPC with respect to the underlying crystalline structure, together with strain induced by the metal gates. As frequently observed in other labs (see e.g. Fig. 5 in ref. [28]), we find that the

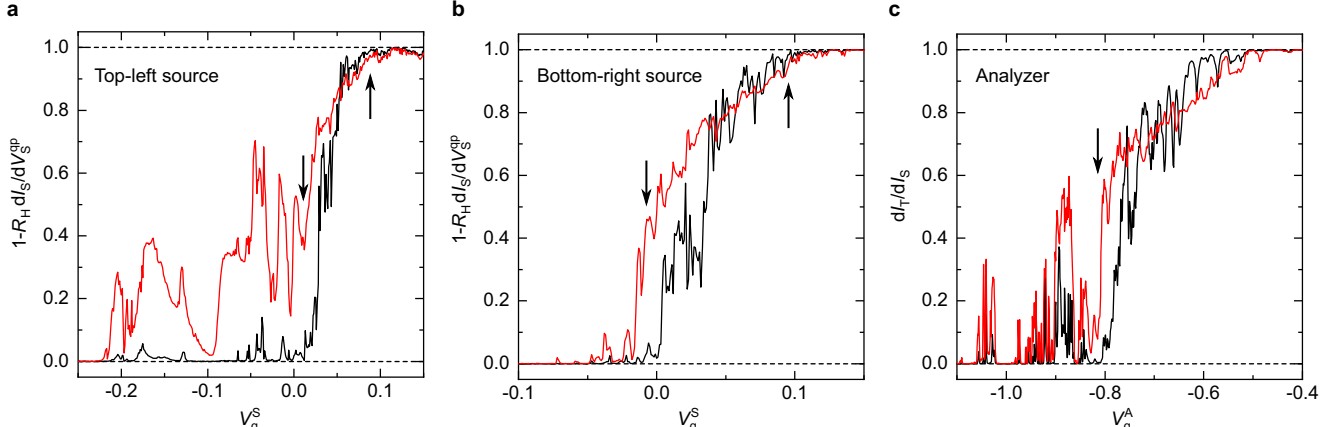

**Fig. 7 | Quantum point contacts vs gate voltage. a b** Differential transmission ratio $1 - R_H dI_S/dV_S^{qp}$ of the top-left (bottom-right) source QPC, as a function of the voltage $V_g^S$ applied to the source QPC gate located the furthest from the analyzer QPC. The black and red continuous lines correspond to measurements in the presence of a dc voltage bias $V_S^{qp} = 0\,\mu V$ and $V_S^{qp} = -43\,\mu V$, respectively. **c** Analyser differential transmission ratio $dI_T/dI_S$ as a function of the gate voltage $V_g^A$ applied to the two gates controlling the analyzer QPC. The black and red continuous lines correspond to measurements in the presence of a direct dc voltage bias $V_S^{qp} = V_S^e = 0\,\mu V$ and $-43\,\mu V$, respectively. The arrows indicate the approximate working points in the SBS (down arrows) and WBS (up arrow) regimes.

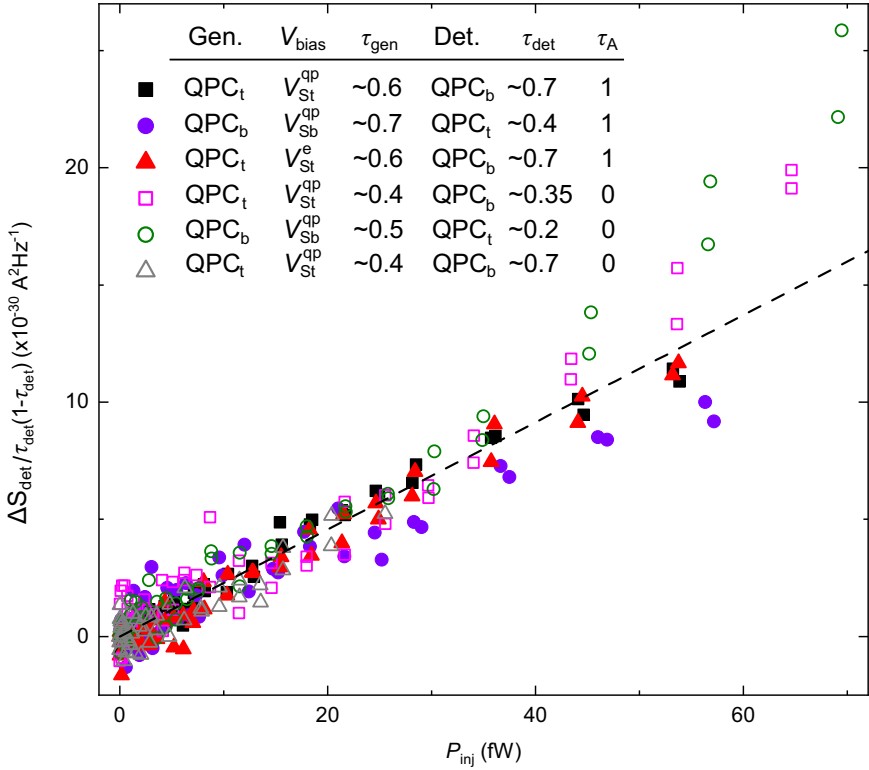

**Fig. 8 | Nonlocal heating.** The normalized noise increase $\Delta S_{det}/(\tau_{det}(1-\tau_{det}))$ emitted from an un-biased `detector' source QPC$_{t(b)}$ of transmission $\tau_{det}$ is plotted as a function of the power $P_{inj}$ injected at a second `generator' source QPC$_{b(t)}$ of transmission $\tau_{gen}$ (t and b indexes stand for the top-left and bottom-right QPCs, respectively). The detector and generator are electrically separated by chirality, and by an incompressible fractional quantum Hall state ($\tau_A = 1$) or a depleted 2DEG ($\tau_A = 0$). The measurements are here performed at $T \simeq 35$ mK, with $P_{inj} = 2V_{bias}\tau_{gen}(1-\tau_{gen})e^2/3h$. The voltage bias $V_{bias}$ is indexed by 'St' or 'Sb' depending on whether it is applied on QPC$_t$ or QPC$_b$. The straight dashed line corresponds to $2.3\,10^{-16}P_{inj}$.

evolution of the transmission with the applied bias changes direction around $\tau \sim 0.8$, thus $\tau$ monotonously decreases with the bias in the WBS regime in contrast with predictions[11,37] (see the diminishing $\tau_S$ with the applied bias in the inset of Fig. 3a where $1 - \tau_S \ll 1$, compared to the increasing $\tau_S$ with the bias in the inset of Fig. 3b where the source QPC is in the SBS regime).

## Absence of a channel substructure along the $\nu = 1/3$ edge

At $\nu = 1/3$, the fractional quantum Hall edge is expected to be composed of a single channel[38]. Although it is also the case at $\nu = 1$, it was previously observed that an additional substructure could emerge[39], possibly due to the smoothness of the edge confinement potential combined with Coulomb interactions. Here we check for the absence of signatures of a substructure along the edge channels connecting the source QPCs to the central, analyzer QPC.

A first indication of a single channel structure is the absence of obvious plateaus at intermediate transmission (see Fig. 7). However, there would be no plateaus if the sub-channels were imperfectly separated at the QPCs. The principle of the substructure test is to compare the transmissions across the analyzer QPC as measured when

a small ac voltage is directly applied or when the impinging ac electrical current first goes through a source QPC (see e.g. refs. [39,40]). In the absence of a substructure and at zero dc bias voltage, the two values must be identical whatever the tunings of the upstream and downstream QPCs. In contrast, a sub-structure robust along the $1.5\,\mu m$ edge path that is associated with any imbalance in the transmission across the source and analyzer QPCs, would result in different values.

At our experimental accuracy, the two signals are systematically found to be identical (data not shown), which corroborates in our device the expected absence of a channel substructure at $\nu = 1/3$.

## Absence of contact noise

A poor ohmic contact quality or other artifacts (electron thermalization in contacts, dc current heating in the resistive parts of the measurement lines…) could result in an unwanted, voltage-dependant noise sometimes refereed to as 'source' noise. Such a noise could spoil the experimental excess noise. Here we checked for any such source noise, and saw that it was absent at our experimental accuracy on the complete range of applied dc voltage bias, both with the device set to have all its QPCs fully open or fully closed.

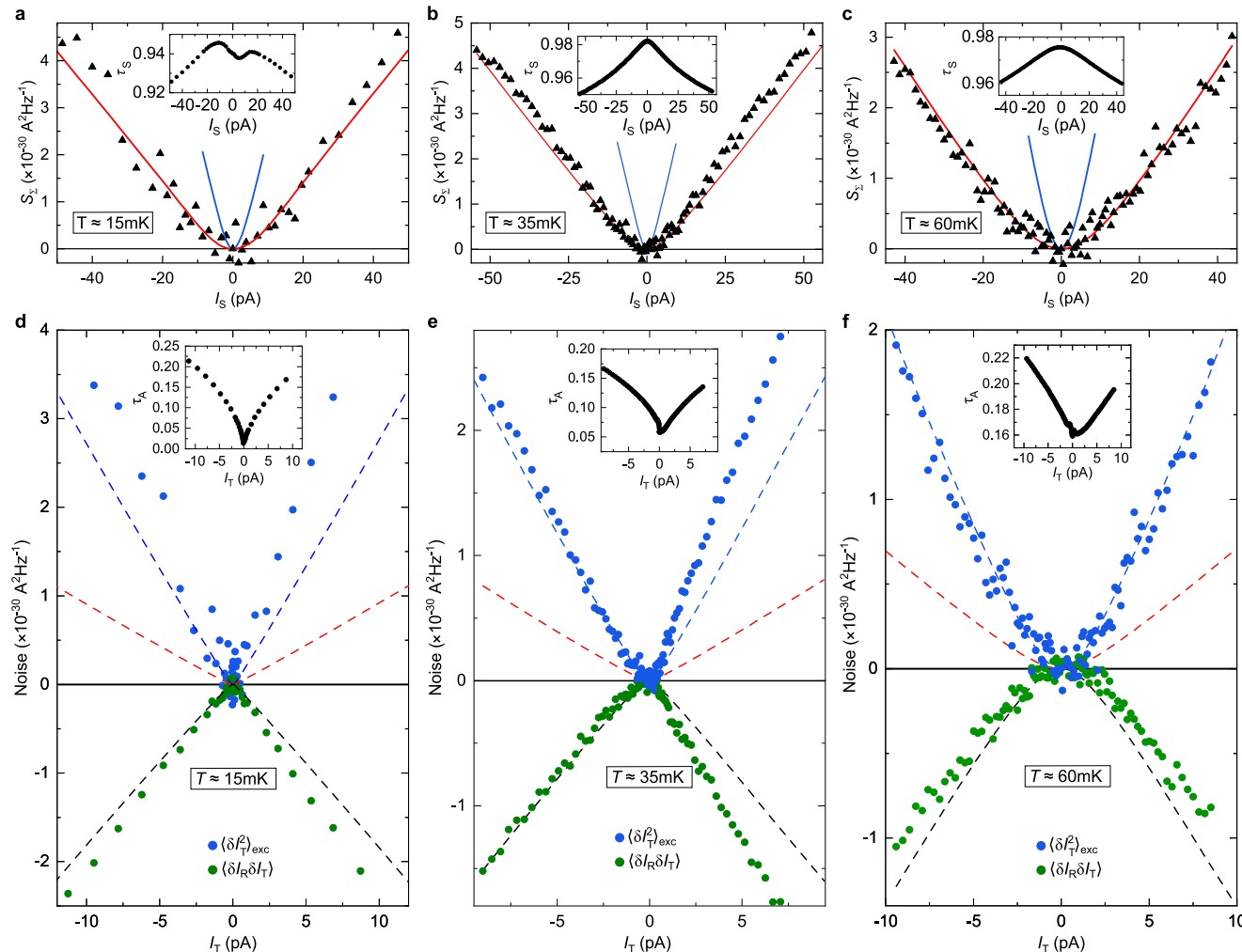

**Fig. 9 | Supplementary Andreev observations, in the source WBS - analyzer SBS regime.** Measurements at the different temperatures $T \approx 15$ mK (**a, d**) and 60 mK (**c, f**) are displayed, as well as measurements at 35 mK obtained using the other (bottom) source QPC located on the opposite side of the analyzer (**b, e**) (see Supplementary Information for a different gate voltage tuning of the device that exhibits a behavior symmetric in the polarity of the bias for both source QPCs). The data in (**a**) and (**d**), in (**b**) and (**e**), and in (**c**) and (**f**) were measured simultaneously. **a, b, c** The top panels show the measured $S_\Sigma$ (black symbols) for the

simultaneous characterization of the source tunneling charge, similarly to Fig. 3a but at different temperatures and with another source QPC (bottom-right in Fig. 1c). The red (blue) lines are the shot noise predictions of Eq. (1) for $e/3$ at the corresponding $T$. **d, e, f** The bottom panels show the auto-correlations in the transmitted current (blue symbols) as well as the cross-correlations between transmitted and reflected current (green symbols), similarly to Fig. 3c. The dashed lines are the predictions of Eq. (5) at the indicated $T$.

## Non-local heating

In a canonical description of the fractional quantum Hall effect at $\nu = 1/3$, the two source QPCs would be completely disconnected from one another and would not be influenced by the downstream analyzer QPC due to the chirality of the edge transport. Whereas the electrical current obeys the predicted chirality, we observe signatures that it is not the case for a small fraction of the heat current. Although discernible (see e.g. the deviations from zero of the black symbols in Fig. 2b), this effect is essentially negligible in the WBS and SBS configurations of present interest. We nevertheless provide here a characterization of this phenomenon.

The non-local heating notably manifests itself as a small noise generated at one of the source QPCs when set to an intermediate transmission ratio, in response to a power injected at the other source QPC. This noise persists even at $\tau_A = 0$, where the two source QPCs are not only separated by the chirality but also by a depleted 2DEG area. This shows that it cannot result from (unexpected) neutral modes going upstream along the edges or through the fractional quantum Hall bulk[41,42]. Instead, we attribute it to a non-local heat transfer involving the long-range Coulomb interaction[39,43].

For the present non-local heating characterization, we set $\tau_A = 0$ or 1, such that the measured electrical noise $\langle \delta I_T^2 \rangle$ and $\langle \delta I_R^2 \rangle$ directly correspond to the noise originating from the corresponding uphill source QPC. A voltage bias is applied to only one of the sources, referred to as the 'generator' here. The signal is the concomitant noise increase measured on the amplification line connected to the other, unbiased

source QPC referred to as the 'detector'. We can generally observe an unexpected increase of the noise from the detector, except if any of the two source QPCs is set to a perfect transmission or reflection, which can be understood as follows. If the transmission ratio across the voltage biased generator QPC is $\tau_{gen} = 0$ or 1, then there is no power locally injected along the edge at the location of this QPC ($\propto \tau_{gen}(1 - \tau_{gen})$, see e.g. supplementary materials in ref. [44]) and the edge channel remains cold downstream from the generator. Consequently, there is no available energy source to heat up the detector and thereby to induce an excess electrical noise. If the transmission ratio across the detector QPC is $\tau_{det} = 0$ or 1, it is now the detector that would not be sensitive to a non-local heating. In particular, there would be no related partition noise (such as the so-called delta-$T$ noise $\propto \tau_{det}(1 - \tau_{det})$, see e.g. refs. [45–47]).

In general, one could expect that such heating would depend on the power $P_{inj}$ locally injected at the generator QPC and that, for a given heating, the induced partition noise generated at the detector would scale as $\tau_{det}(1 - \tau_{det})$. Accordingly, we show in Fig. 8 the detector excess noise normalized by $\tau_{det}(1 - \tau_{det})$ as a function of the power injected at the generator, measured at a temperature $T \simeq 35$ mK. In this representation, the data obtained in different configurations fall on top of each other. It mostly does not depend on which of the source QPCs plays the role of the generator or the detector, on which dc voltage is used to bias the source, on whether $\tau_A = 0$ or 1, or on the values of $\tau_{det}$ and $\tau_{gen}$. Based on this observation and interpretation, it is possible to estimate the impact of such non-local heating assuming a non-chiral noise on an unbiased QPC to be $\sim P_{inj} \tau_{det}(1 - \tau_{det}) \times 2.3 \, 10^{-16} \, \text{A}^2/\text{Hz}$

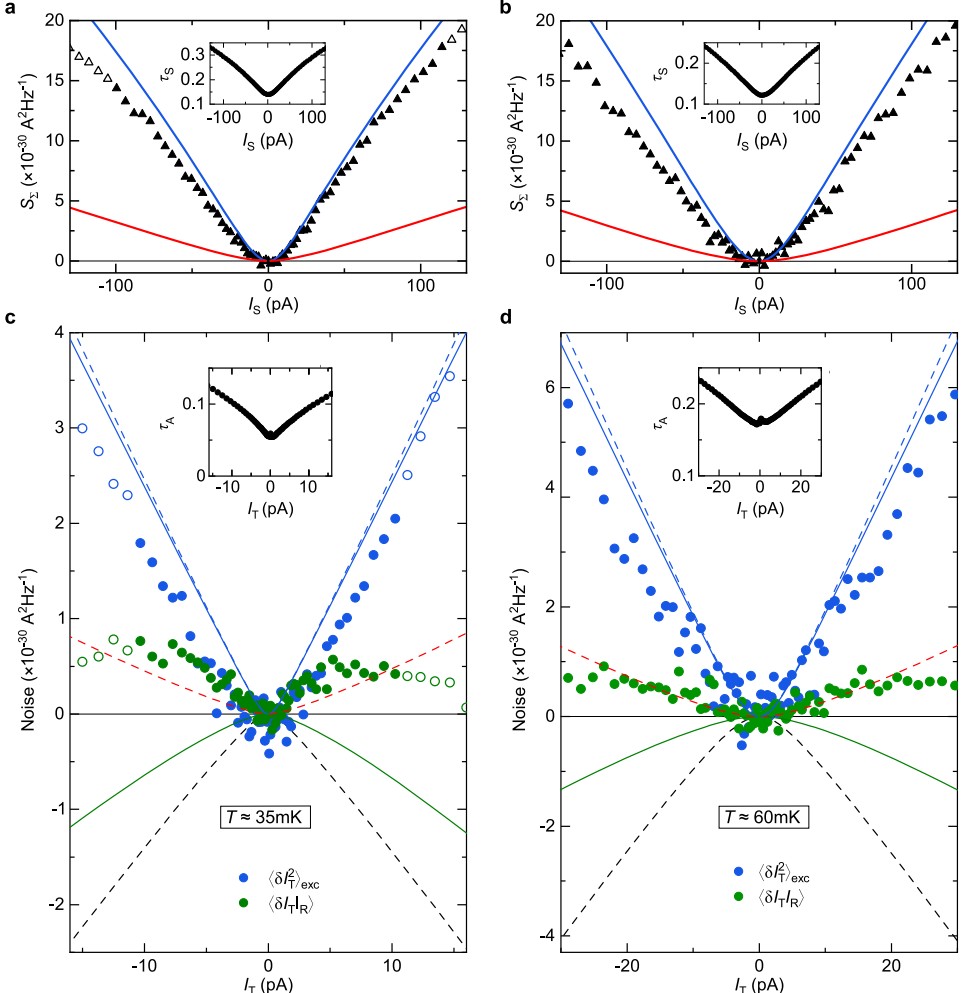

**Fig. 10 | Supplementary observations in the source SBS - analyzer SBS regime. a, b, c, d** The displayed data (symbols) corroborate the observations shown in Fig. 3b,d for distinct device tunings, and also at the higher temperature $T \approx 60$ mK (**b, d**). The data in (**a**) and (**c**), and in (**b**) and (**d**) were measured simultaneously.

(dashed line). Note that such heating should also take place between the analyzer and the upstream sources, which corresponds to the small increase of $S_\Sigma$ at high bias in Fig. 2b. In that specific case, a (unexpected) neutral counter-propagating heat flow could also take place, in principle, however the smallness of the heating signal rules out a substantial additional contribution to the above non-local heating. Importantly, in the main configurations with the sources set to or near a transmission of 0 or 1, we typically expect a negligible impact of only a few percent or less on the auto- and cross-correlations of interest. Moreover, when the detector QPC of tunneling charge $e^*$ is a quasiparticle source itself voltage biased at $V_{bias}$, we expect that the noise resulting from a non-local heating vanishes (in the limit of a small heating with respect to $e^* V_{bias}/k_B$, see Eq. (1)).

### Fit expressions

Here we provide the specific expressions used to fit the auto/cross-correlation data in the different configurations, when not explicitly given in the main text.

In the source-analyzer configurations shown in Fig. 3c, d and Fig. 4 (as well as in Fig. 9d, e, f and Fig. 10c, d in Methods), the different slopes of the dashed lines are associated with the thermal rounding of the source QPC. Explicitly, the displayed dashed lines correspond to:

$$S = 2e^* I_T \left[ \coth \frac{e_S^* V}{2k_B T} - \frac{2k_B T}{e_S^* V} \right], \tag{5}$$

with $e_S^* V = (e/3)V_S^{qp}$ for Fig. 3c and Fig. 4 (as well as Fig. 9d, e, f in Methods), $e_S^* V = e V_S^e$ for Fig. 3d (as well as Fig. 10c, d in Methods), and the prefactor $e^* = e$, $e/3$ and $-2e/3$ for the blue, red and black dashed lines, respectively.

The noninteracting electron expressions for a source-analyzer configuration, which are displayed as continuous lines in Fig. 3d (as well as in Fig. 10c, d in Methods), are provided below. The auto-correlations of the transmitted current (continuous blue line) is given by:

$$\langle \delta I_T^2 \rangle_{exc} = 2e I_T (1 - \tau_A \tau_S) \left[ \coth \frac{e V_S^e}{2k_B T} - \frac{2k_B T}{e V_S^e} \right], \tag{6}$$

and the cross-correlations (continuous green line) is given by:

$$\langle \delta I_R \delta I_T \rangle = -2e I_T (1 - \tau_A) \tau_S \left[ \coth \frac{e V_S^e}{2k_B T} - \frac{2k_B T}{e V_S^e} \right]. \tag{7}$$

### Andreev observations for different temperatures and tunings

The robustness of our observations is ascertained by repeating the measurements at different temperatures, by using a different QPC for the source, and by using different tunings of the source and analyzer QPCs.

Figure 9 shows such additional measurements in the Andreev configuration of a source in the WBS regime and an analyzer in the SBS regime. The main changes compared to Fig. 3c are the additional temperatures of $T \approx 15$ mK and 60 mK in Fig. 9a,d,c,f, and that a different QPC (located on the opposite side of the analyzer) is used for the source in Fig. 9b,e (see Supplementary Information for further data in the Andreev configuration). Note that at the lowest 15 mK temperature, the very fast increase with direct bias voltage of the transmission $\tau_A$ across the analyzer set in the SBS regime makes it difficult to unambiguously ascertain, separately, its $1e$ characteristic tunneling charge (data not shown).

Figure 10 shows additional measurements when the source and analyzer are both set in the SBS regime. A similar signal as in Fig. 3b,d is observed for a different tuning of the device and at the higher temperature $T \approx 60$ mK.

## Data availability

The data that support the findings of this study are available from the corresponding authors upon reasonable request. The raw measurements used in this article, the codes to analyze these measurements, and the data plotted in the figures are available via Zenodo at https://doi.org/10.5281/zenodo.10091819.

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

## Acknowledgements

This work was supported by the European Research Council (ERC-2020-SyG-951451), the European Union's Horizon 2020 research and innovation programme (Marie Skłodowska-Curie grant agreement 945298-ParisRegionFP), the French National Research Agency (ANR-16-CE30-0010-01 and ANR-18-CE47-0014-01) and the French RENATECH network. The authors thank C. Mora and H.-S. Sim for illuminating discussions.

## Author contributions

O.M. and P.G. contributed equally to this work; O.M. and P.G. performed the experiment and analyzed the data with inputs from A.Aa., A.An., C.P. and F.P.; A.C. and U.G. grew the 2DEG; A.Aa, F.P., O.M. and P.G. fabricated the sample; Y.J. fabricated the HEMT used in the cryogenic noise amplifiers; F.P., O.M. and P.G. wrote the manuscript with contributions from A.Aa., A.An., C.P. and U.G.; A.An. and F.P. led the project.

## Competing interests

The authors declare no competing interests.
