## [Peer Review File · Nature Communications]

Quasiparticle Andreev scattering in the $\nu = 1/3$ fractional quantum Hall regimeREVIEWER COMMENTS

Reviewer #1 (Remarks to the Author):

The authors report the Andreev-like reflection in a $1/3$ fractional quantum Hall edge; a $1/3$ quasiparticle is converted into a " $-2e/3$ " hole and " e " quasielectron at a qpc. As evidence of the Andreev-like reflection, they observed the negative cross correlation between the currents measured at two different drains, and the measured values of the cross correlation match well with the theoretical prediction. While the Andreev-like reflection is interesting and the analysis they used sound reasonable, I have a few questions and comments:

1. First, I would like to appreciate for showing all the characteristics of the qpcs they have used. In the inset of Fig. 2a, τ_S slightly decreases upon increasing the bias voltage. I think the simple Luttinger liquid theory expects the opposite since the $1/3$ quasi-particle tunneling is RG relevant. Is it the finite-temperature effect that masks the low-voltage behavior? In this regard, it would be instructive if you show the few data points, measured at different gate voltage to tune the qpc.

2. In the inset of Fig. 2c, the data of the transmission seems rather noisy in the small bias voltage regime. What really happens in the regime? One may expect that it would be monotonous by temperature.

3. An intriguing feature is the positive cross correlation for the dilute beam in the SBS regime. What I would like to point out is that one cannot neglect the noise of the dilute beam in the SBS regime. τ_A is sufficiently large ($>.1$) even in the small I_T so that the noise of the dilute beam may contribute a sizeable signal to the cross correlation. Furthermore, this contribution is positive because $\langle \delta I_s \delta I_T \rangle > 0$, where I_s is the impinging current. In this sense, it would be good to show $S_{\{\Sigma\}}$ also for Fig. 2(c) and (d) like Fig.1b if available.

4. Finally, it would be good to discuss the effect of the diluted beam. My understanding is that while the impinging full-beam allows the triplet of quasiparticles to tunnel through a qpc, quasiparticles in the diluted beam are not able to find their own triplet so that only Andreev-like reflection is allowed. Does it right?

Reviewer #2 (Remarks to the Author):

The present experimental work investigates the complexity of fractional quasiparticles with a very simple device. While the work pushed me to think diligently about the explanation, I do have some concern regarding the results. Here is a list of the few that mainly confused me

1. The fig. 2c shows noise at current of 5pA with value $>1e-30$ while I expect $>1e-31$ through simple calculation. Also, the data is clean with no noise of the noise, I think it will be really help to see the real data before averaging. This is in fact related to all data in the range of near zero current, while the authors get 'a single nice point', should have been averaged for super long time

2. I also noticed that there is almost no temperature rounding around 1pA, leading to energy of $e/3$ particles of ~ 25 meV, while the temperature of the dilution is 35 mK which must be reflected in the data near zero bias, and the broadening corresponding to 35 mK is about 2 μ eV!

3. While the transmission changes by a factor of > 3 , the noise sits on a straight line? Which transmission has been used for the fitting curves?

4. Also, it is not clear where is the claim of reflected $(-2/3)e$ being based on negative cross-correlation. It is widely known that the negative cross-correlation of $T - R$, which results from a full beam coming at a QPC – and never found a similar claim of holes are being reflected from the QPC. The authors might see the holes, but this negative cross-correlation should be explained here in more details.

5. The ref. 14 showed also AR, while from filling $1/3$ to 1 (but not pinched qpc), the idea was published that didn't see this effect, hence that must be discussed for clarity.

Finally, some side comments, the paper is hard to read, must be made easier to follow.

Also, the authors talk about ref. 15 that didn't see AR due to long-distance, but why so? The long-distance leads to equilibration and definitely negative cross-correlation. The authors also said, ref. 16 mitigates ref. 15, I am not sure I understood how?

Reviewer #3 (Remarks to the Author):

The authors of "Quasiparticle Andreev scattering in the $\nu=1/3$ fractional quantum Hall regime" created a scenario where a dilute beam of Laughlin quasiparticles is impinging onto a barrier which transmit only electrons. This reflection was coined in the past 'Andreev' in similarity to Andreev reflection in superconductor-normal junctions. The authors have realized this scenario in a high-mobility GaAs/AlGaAs heterostructure at the fractional quantum Hall effect. A diluted beam of Laughlin quasiparticles was attained by applying a current impinging a source quantum point contact in high transmission while electron tunneling was enforced by using a second analyzer QPC in a low transmission.

The authors/group are known for shot-noise measurements in the quantum Hall effect and indeed their measurements technique and analysis look proper and solid. Below I have suggested several points which I feel will help the authors to transmit their measurements in a more coherent manner for the reader.

My main criticism in this work is that I do not see the main point the authors are trying to make and the new physics it is bringing. In general, when thinking on an $e/3$ quasiparticle which tunnel through vacuum one expects to see a tunneling of e which naively is achieved via a bunching of 3 quasiparticles. Unlike 'regular' Andreev reflection all this process occurs at the same energy, and it is hence relatively understood. The authors in my opinion need to stress the novelty in their work. Again, the measurements and analysis the authors are doing here are difficult and interesting however a considerable amount of work should direct these findings in conveying a new physics finding.

Knowing the group work I suggest the authors make a major revision which concentrate on the physics interpretation of their work stressing the novelty in it. Perhaps something that can teach us new insights in the direction of edge reconstruction?

I would also add as a side note that following a relatively recent paper by Feve (ref. 25) on a similar structure (different regime), would be interesting to hear the report of the authors measurements in that direction. The impact of new insights in that direction I feel would be valuable to the community.

A few minor comments which might help the reader – mainly on figures to ease the reader – The title of the paper relates the physics to Andreev reflection. I understand the choice of the authors and the relative connection. It was made before as the authors state in ref. 14, there it was an interface between fractional and an integer filling. Here it's the same filling from both sides and only the tunneling constriction enables electron transport. I believe it would help the reader allot to have some introduction figure where there is more of a schematic representation of the

experiment. Showing the difference from regular AR and showing the important physics here.

- Inset in figure 1a is confusing it gives the feeling that it is ff $1/3$ on one side and ff 1 on the other. Better to make under inset or make filling factors on the SEM image.
- Would be good to state the relative transmission of the two qpcs on the SEM image. When someone is looking at it they can understand immediately the scenario.
- I believe figure 1b should be more self-explanatory. Maybe writing 'e' or 'e/3' on the different. Figure is very hard to follow even for someone from the field and this is the first figure in the paper.
- Panel should be above/below so the reader can correlate immediately the transmission to noise.
- The word bunching doesn't appear even once in the paper where it is actually a very common way in the community to think of this process – I believe the authors should address this point.

We would like to express our appreciation for the reviewers' in depth and constructive remarks. We believe the manuscript is now vastly improved in clarity, thanks to the reviewers' suggestions, comments and questions.

Reviewer #1

The authors report the Andreev-like reflection in a $1/3$ fractional quantum Hall edge; a $1/3$ quasiparticle is converted into a " $-2e/3$ " hole and " e " quasielectron at a qpc. As evidence of the Andreev-like reflection, they observed the negative cross correlation between the currents measured at two different drains, and the measured values of the cross correlation match well with the theoretical prediction. While the Andreev-like reflection is interesting and the analysis they used sound reasonable, I have a few questions and comments:

1. First, I would like to appreciate for showing all the characteristics of the qpcs they have used. In the inset of Fig. 2a, τ_S slightly decreases upon increasing the bias voltage. I think the simple Luttinger liquid theory expects the opposite since the $1/3$ quasi-particle tunneling is RG relevant. Is it the finite-temperature effect that masks the low-voltage behavior? In this regard, it would be instructive if you show the few data points, measured at different gate voltage to tune the qpc.

[Reply]

As noticed by the reviewer, all our QPCs display at high transmission a reduction of τ away from the ballistic limit ($\tau=1$) as the voltage bias is increased. This is in stark contrast with a simple Luttinger description, and not a finite temperature effect (the maximum voltage bias in the corresponding figure 3a is of $85\mu\text{V}$, much larger than $kT/(e/3)\sim 9\mu\text{V}$).

Such a behavior, although unexplained, is frequently observed experimentally in the fractional quantum Hall regime. See for instance Fig. 5 in [Heiblum and Feldman, Int J Mod Phys A 35, (2020)] where τ is decreasing with the bias voltage above a transmission of ~ 0.8 , in "drastic disagreement in the weak back scattering regime with the chiral Luttinger liquid prediction" (quote from caption).

The reviewer can appreciate the robustness of this behavior for our QPCs in Fig. 7 (previous Extended Data Fig. 3), where gate voltage sweeps of the differential transmissions of the three QPCs are displayed at zero and high ($-43\mu\text{V}$) bias voltage. The high bias transmission (red) is lower at $\tau\sim 0.8$ in contrast with a simple Luttinger model (except for some fast oscillations interpreted as resonances).

In response to the reviewer, we have added a note in the text pointing out this unorthodox behavior and also the related Fig. 7 in Methods as well as the above mentioned reference (new Ref. 26).

[Reviewer #1]

2. In the inset of Fig. 2c, the data of the transmission seems rather noisy in the small bias voltage regime. What really happens in the regime? One may expect that it would be monotonous by temperature.

[Reply]

We do indeed expect a monotonous increase, and the data in Fig. 3c (previous Fig. 2c) are consistent with this picture at our experimental accuracy.

The transmission data on $\tau_A = I_T/I_S$ at small I_T is more noisy when measured in-situ, using the small quasiparticle current I_S emitted by the source, simply because there is little signal to determine this ratio. Compared to a direct voltage bias, such as for the determination of τ_A in Fig. 2a or for the determination of the transmission across the source QPC, the probed currents are reduced by a factor $(1-\tau_S) \sim 0.02$.

We now point out explicitly that the increased noise results from a reduced signal.

[Reviewer #1]

3. An intriguing feature is the positive cross correlation for the dilute beam in the SBS regime.

What I would like to point out is that one cannot neglect the noise of the dilute beam in the SBS regime. τ_A is sufficiently large ($>.1$) even in the small I_T so that the noise of the dilute beam may contribute a sizeable signal to the cross correlation.

Furthermore, this contribution is positive because $\langle \delta I_S \delta I_T \rangle > 0$, where I_s is the impinging current.

In this sense, it would be good to show S_{Σ} also for Fig. 2(c) and (d) like Fig.1b if available.

[Reply]

We agree with the reviewer that, beyond the asymptotic limit $\tau_A \ll 1$, there will be some deviation to a zero cross correlations even in a non-interacting fermions model. We also agree that positive cross-correlations $\langle \delta I_S \delta I_T \rangle$ are expected, as illustrated most strikingly by the opposite ballistic limit $\tau_A = 1$ where $\delta I_T = \delta I_S$.

However, we find that the cross correlations between transmitted and reflected signal $\langle \delta I_R \delta I_T \rangle$ should remain negative, vanishing in the ballistic limit $\tau_A = 1$ whatever τ_S and also vanishing in the tunneling limit $\tau_S \rightarrow 0$ whatever τ_A . The full theoretical expression in the noninteracting case, valid for any τ_A and τ_S , is provided in Methods, Eq. 7, and corresponds to the continuous green line in Fig. 3d (previous Fig. 2d).

In response to the reviewer we have added in the caption of the figure a reference to this equation and explicitly indicate that it is valid for any τ_A, S . The corresponding S_{Σ} is displayed on the panels above Fig. 3c,d (old Fig. 2c,d).

[Reviewer #1]

4. Finally, it would be good to discuss the effect of the diluted beam.

My understanding is that while the impinging full-beam allows the triplet of quasiparticles to tunnel through a qpc,

quasiparticles in the diluted beam are not able to find their own triplet so that only Andreev-like reflection is allowed.

Does it right?

[Reply]

Indeed. With a dilute beam of well-separated incoming quasiparticles of charge $e/3$ arriving at the analyzer QPC, the tunneling charge e is not readily available (no bunching mechanism of several nearby quasiparticles is possible). In principle, a possibility could have been that the transport is then simply blocked ($\tau_A = 0$) or that a normally negligible mechanism involving the transmission of charges $e/3$ becomes dominant. However it turns out, as predicted theoretically and observed here, that the

missing charge can be sucked in from the incident channel leaving a hole of $-2e/3$. This mechanism could be described in terms of bunching of $e/3$ quasiparticles together with the excitations of two additional quasiparticle-quasihole pairs. In the same way as a quasidelectron can be pictured as three bunched $e/3$ quasiparticles, the reflected $-2e/3$ quasihole can also be seen as two co-propagating $-e/3$ quasiholes.

We have improved the description of the Andreev mechanism in the new version of the manuscript (page 1 and new Fig. 1a,b), and now provide a graphical picture of this dual representation in the new Fig. 1a.

Reviewer #2

The present experimental work investigates the complexity of fractional quasiparticles with a very simple device. While the work pushed me to think diligently about the explanation, I do have some concern regarding the results. Here is a list of the few that mainly confused me

1. The fig. 2c shows noise at current of 5pA with value $>1e-30$ while I expect $>1e-31$ through simple calculation. Also, the data is clean with no noise of the noise, I think it will be really help to see the real data before averaging.

This is in fact related to all data in the range of near zero current, while the authors get 'a single nice point', should have been averaged for super long time

[Reply]

Addressing the second point first, the vertical scatter of the noise signal in Fig. 3c (previous Fig. 2c) is of about $10^{-31} \text{ A}^2/\text{Hz}$. Such a resolution is good, but not exceptional for the standards of our team (a resolution of $5 \cdot 10^{-32} \text{ A}^2/\text{Hz}$ was achieved at $\nu=2$ despite the reduced experimental sensitivity due to the lower quantum Hall resistance, see [Nat. Phys. 14, 145 (2018)]) and also with respect to experimental data from other teams (see e.g. Fig. 5b,c in [Heiblum and Feldman, Int J Mod Phys A 35, (2020)]).

In practice, we make several sweeps each with a noise integration of 10s per point. For Fig. 3c (3d), we averaged 12 (2) sweeps, resulting in an effective integration time of 120s (20s) per point. Note that the vertical scatter between data points closely corresponds to the standard error of the mean calculated from the averaged ensemble of points. For instance, the vertical scatter in Fig. 3d is of about $3 \cdot 10^{-31} \text{ A}^2/\text{Hz}$, consistent with the expected increase of the noise as $\sqrt{120/20}=2.5$ with respect to the data in panel (c).

Although it is difficult to distinguish because of the density of nearby points, the excess autocorrelation (blue symbols) is exactly zero only at $I_T=0$. This is by construction, because the displayed excess autocorrelations are defined as the difference with respect to the zero bias noise.

We have double-checked our calculations. A simple $2 e I_T$ at $I_T=5\text{pA}$ gives $1.6 \cdot 10^{-30} \text{ A}^2/\text{Hz}$, very close to the dashed blue line (slightly higher because of the thermal rounding accounted for in the prediction shown as a dashed blue line). Note that the value would be smaller for the tunneling of $e/3$ quasiparticles for which $2 (e/3) I_T \sim 5 \cdot 10^{-31} \text{ A}^2/\text{Hz}$ at 5pA (close to the dashed red line).

In response to the reviewer, we have added in Methods (see end of 'Experimental setup') more precisions on the noise averaging procedure and now explicitly point out that the vertical scatter closely adequately indicate the standard error of the mean.

[Reviewer #2]

2. I also noticed that there is almost no temperature rounding around 1pA, leading to energy of $e/3$ particles of ~ 25 neV, while the temperature of the dilution is 35 mK which must be reflected in the data near zero bias, and the broadening corresponding to 35 mK is about 2 μ eV!

[Reply]

We assume that the reviewer speaks about the noise data with the source QPC tuned at $e/3$ as shown in Fig. 3a (previous Fig. 2a, the only $e/3$ noise data in the main text).

The corresponding finite temperature prediction calculated from Eq. 1, using the separately calibrated temperature of 35mK and $e^*=e/3$, is displayed as a red continuous line in Fig. 3a. As a quick order of magnitude check, a rounding in I_S of about 3pA corresponds for $\tau_S \sim 0.98$ to an applied voltage of $I_S \cdot (3 e^2/h)/(1-\tau_S) \sim 12\mu V \sim (e/3) k_B 45$ mK. This differs from the reviewer's value of 25neV, which may have been obtained by simply taking the product of the quantum of resistance h/e^2 and 1pA. However I_S is not the incident current on the source QPC, but the back-scattered current from this QPC (the incident current on the source is $I_S/(1-\tau_S) = V_S^{qp}/(3h/e^2)$).

In response to the reviewer, we now indicate also in the caption of Fig. 3a that the continuous red line is the prediction of Eq. 1 for $T=35$ mK. We have also improved the manuscript to better clarify the connection between I_S , V_S^{qp} and τ_S . In particular, the transmission τ_S is now graphically represented in Fig. 1c and a specific discussion was added in the caption. Furthermore, graphical representations of the device setting and biasing are now displayed as insets in Figs. 2,3,4

In the eventuality that the reviewer was mentioning the 'thermal rounding' of the main cross-correlations signal in Fig. 3c, we point out that it is even narrower because, in this case (as opposed to the data in Fig. 2b or Fig. 3a), the analyzer QPC is not directly voltage biased but fed from an upstream voltage biased source. It is thus the voltage applied to the upstream source that should be compared with the temperature, in good agreement with the displayed continuous line (obtained from Eq. 5 with $T=35$ mK as detailed in Methods, section 'Fit expressions').

[Reviewer #2]

3. While the transmission changes by a factor of > 3 , the noise sits on a straight line? Which transmission has been used for the fitting curves?

[Reply]

The precise definition of the transmission (differential vs dc) is indeed important when τ is not constant. Accordingly, we define all τ as a ratio of transmitted over incident dc currents. Thus $2e^* \tau(1-\tau)V/(h/e^2/\nu) = 2 e^* I_{\text{transmitted}} (1-\tau) = 2e^* I_{\text{backscattered}} \tau$, and one can see that the robust Poissonian expression $2 e^* I$ holds even for strong relative changes of τ and $1-\tau$ as long as the QPC remains in the WBS or SBS limits.

In response to the reviewer, the definition of τ as a dc current ratio is now recalled also below Eq. 1.

[Reviewer #2]

4. Also, it is not clear where is the claim of reflected $(-2/3)e$ being based on negative cross-correlation. It is widely known that the negative cross-correlation of $T - R$, which results from a full beam coming at a QPC – and never found a similar claim of holes are being reflected from the QPC. The authors might see the holes, but this negative cross-correlation should be explained here in more details.

[Reply]

Negative cross correlations do occur under a direct voltage bias, such as observed in Fig. 2b (previous Fig. 1b). However, in this direct bias case the cross-correlations are opposite to the excess auto-correlations, with the same magnitude (as in Fig. 2b), whatever the transmitted quasiparticle. This results from the fact that the quasiparticles (or more generally the current fluctuations) are generated at the same QPC together with local current conservation.

In the non-interacting scattering picture for a dilute incident beam of quasiparticles, each quasiparticle would be either transmitted or reflected corresponding to a vanishing cross-correlations Fano factor (dilute limit for the source, see e.g. Eq. 7 with $\tau_S \ll 1$ for dilute quasielectrons obtained applying a bias voltage $V_{qp}e$).

In the present case, we find that the ratio between auto correlations on the tunneling current across the analyzer QPC $\langle \delta I_T^2 \rangle$ and the cross-correlations $\langle \delta I_T \delta I_R \rangle$ is neither -1 nor 0 but $-2/3$. This shows that for each fluctuation in the tunneling current δI_T there is a correlated fluctuations $\delta I_R = -2/3 \delta I_T$. As we find a tunneling charge of $1e$, each of the tunneling quasielectrons are accompanied by a reflected hole of charge $-2/3 * (1e)$.

In response to the reviewer, we have clarified below Eq. 2 the connection between the cross-correlations amplitude and the value of the reflected charge.

[Reviewer #2]

5. The ref. 14 showed also AR, while from filling $1/3$ to 1 (but not pinched qpc), the idea was published that didn't see this effect, hence that must be discussed for clarity.

[Reply]

This reference investigates a different kind of Andreev mechanism, taking place at an interface between two different quantum Hall states. The authors found some conductance evidences for this kind of Andreev reflection. We are not aware of any subsequent article contesting these evidences, and it is not our intention to cast any doubt on their validity if that is what the reviewer is asking. Possibly the reviewer has in mind the sentence previously in the abstract “However, pioneer experiments were not able to find direct evidences”, which concerned the present kind of Andreev mechanism with single incident quasiparticles (whereas the [14] deals with a mechanism occurring at a voltage biased interface) but we were actually referring to the pioneer experiments [15,16] (references are not allowed in the abstract).

In response to the reviewer the above sentence has now been removed (also to shorten the abstract).

As a side note, it can be seen from the peer review file of [14] (https://static-content.springer.com/esm/art%3A10.1038%2Fs41467-021-23160-6/MediaObjects/41467_2021_23160_MOESM2_ESM.pdf) that a direct characterization of the charges in the $\nu=1$ and $\nu=1/3$ sides (as presently done with the noise in the present Andreev-like mechanism) would have been considered a definite claim.

[Reviewer #2]

Finally, some side comments, the paper is hard to read, must be made easier to follow.

[Reply]

Numerous modifications were made to the manuscript and figures to make it clearer and easier to read.

[Reviewer #2]

Also, the authors talk about ref. 15 that didn't see AR due to long-distance, but why so? The long-distance leads to equilibration and definitely negative cross-correlation.

[Reply]

Our only message is that the extremely long distance between source and analyzer (with respect to e.g. phase coherence length normally $< \sim 10 \mu\text{m}$) in [15,16] could play a role, but we are not committing into a specific explanation.

The reviewer suggests that the relaxation process would give rise to an effective voltage bias (at base T) corresponding to the current emitted by the source. The application of a voltage results in a Fano factor on the autocorrelations of the transmitted current corresponding to a tunneling charge of e (and negative cross-correlations although not measured in [15,16]). This contrasts with the finding of a tunneling charge of $e/3$ in the dilute limit in [15], which may reflect the fact the effective voltage on the second (analyzer) QPC is not constant but fluctuates in the probed MHz range with the current fluctuations corresponding to the emission of $e/3$ quasiparticles from the far away source.

Given the speculative nature of these remarks, we prefer to hold on to our simple original statement.

[Reviewer #2]

The authors also said, ref. 16 mitigates ref. 15, I am not sure I understood how?

[Reply]

Ref. 15 found that the Fano factor of the autocorrelations of the current tunneling across the analyzer QPC in the SBS regime correspond in the dilute beam limit to a charge $e/3$ (in contrast with Kane prediction of an Andreev-like scattering). However, it was seen in the subsequent work Ref. 16 that reducing the temperature could increase the Fano factor/tunneling charge above $e/3$ (depending on the dilution), thus mitigating the generic character of the observations in [15]. Note that the most revealing cross-correlations were not measured in these early works.

In response to the reviewer we have clarified the corresponding sentence.

Reviewer #3

The authors of "Quasiparticle Andreev scattering in the $\nu=1/3$ fractional quantum Hall regime" created a scenario where a dilute beam of Laughlin quasiparticles is impinging onto a barrier which transmit only electrons. This reflection was coined in the past 'Andreev' in similarity to Andreev reflection in superconductor-normal junctions. The authors have realized this scenario in a high-mobility GaAs/AlGaAs heterostructure at the fractional quantum Hall effect. A diluted beam of Laughlin

quasiparticles was attained by applying a current impinging a source quantum point contact in high transmission while electron tunneling was enforced by using a second analyzer QPC in a low transmission.

The authors/group are known for shot-noise measurements in the quantum Hall effect and indeed their measurements technique and analysis look proper and solid. Below I have suggested several points which I feel will help the authors to transmit their measurements in a more coherent manner for the reader.

My main criticism in this work is that I do not see the main point the authors are trying to make and the new physics it is bringing. In general, when thinking on an $e/3$ quasiparticle which tunnel through vacuum one expects to see a tunneling of e which naively is achieved via a bunching of 3 quasiparticles. Unlike 'regular' Andreev reflection all this process occurs at the same energy, and it is hence relatively understood. The authors in my opinion need to stress the novelty in their work. Again, the measurements and analysis the authors are doing here are difficult and interesting however a considerable amount of work should direct these findings in conveying a new physics finding.

Knowing the group work I suggest the authors make a major revision which concentrate on the physics interpretation of their work stressing the novelty in it. Perhaps something that can teach us new insights in the direction of edge reconstruction?

[Reply]

We have rewritten our discussion in the first page of the article to better clarify the main point and new physics in the presently observed Andreev-like reflection.

As indicated by the reviewer, the transmitted $1e$ quasielectron can also naively be seen as three $e/3$ quasiparticles bunched together, both in the present Andreev-like process and for the direct voltage bias of a QPC in the SBS regime. However, whereas these quasiparticles are readily available for pickup in the presence of a direct voltage bias, it is not the case with a single incident $e/3$ quasiparticle. In principle, this could have led to a blocked transport or to the emergence of normally negligible transfer of $e/3$ quasiparticles. What was predicted and here observed is that an additional charge of $2e/3$ is sucked in from the incident channel, leaving a $-2e/3$ hole behind. The main point of our work is thus to demonstrate the emergence of this novel Andreev-like mechanism for single incident quasiparticles.

In what respect does it involve new physics?

In contrast with the standard scattering picture of individual particles, neither the number of particles nor their nature are conserved in the Andreev-like process.

Compared to the bunch transfer of three $e/3$ quasiparticles across a voltage biased QPC, which was pointed out by the reviewer to occur at the same energy making it relatively understood, here the energy of the incident $e/3$ quasiparticle redistributes between transferred quasielectron and reflected quasihole. The energy of the incident quasiparticle thus differs from the individual energy of the outgoing quasiparticles, which stresses the different physics at work.

[Reviewer #3]

I would also add as a side note that following a relatively recent paper by Feve (ref. 25) on a similar structure (different regime), would be interesting to hear the report of the authors measurements in that direction. The impact of new insights in that direction I feel would be valuable to the community.

[Reply]

We agree with the reviewer. This is however a long and different story. A dedicated article is being prepared on this specific topic.

[Reviewer #3]

A few minor comments which might help the reader – mainly on figures to ease the reader

- The title of the paper relates the physics to Andreev reflection. I understand the choice of the authors and the relative connection. It was made before as the authors state in ref. 14, there it was an interface between fractional and an integer filling. Here it's the same filling from both sides and only the tunneling constriction enables electron transport. I believe it would help the reader allot to have some introduction figure where there is more of a schematic representation of the experiment. Showing the difference from regular AR and showing the important physics here.

- Inset in figure 1a is confusing it gives the feeling that it is $\nu = 1/3$ on one side and $\nu = 1$ on the other. Better to make under inset or make filling factors on the SEM image.

[Reply]

We have improved our presentation thanks to the reviewer's advice. The previous Fig. 1 is now split into two figures. The panel (b) of the new Fig. 1 provides a novel schematic representation of the experiment when tuned to show the Andreev-like mechanism. The new panel (a) illustrates the Andreev mechanism, also showing the dual picture of the quasielectron and $-2/3$ reflected quasihole as bunched $e/3$ quasiparticle and quasiholes.

[Reviewer #3]

- Would be good to state the relative transmission of the two qpcs on the SEM image. When someone is looking at it they can understand immediately the scenario.

[Reply]

We hope that the schematic representation in the new Fig. 1b addresses this request of the reviewer (see also text in caption). In the spirit of the reviewer's suggestion, we added new schematics in Fig. 3c,d, in Fig.2b and in Fig. 4 to illustrate the different device tunings.

Regarding Fig. 1c with the SEM image, we prefer to keep the picture neutral as the relative transmissions of the QPCs depend on the device tuning (we also show some data with the source set in SBS as counter-point to the main Andreev configuration). Note however that, in the spirit of the reviewer's request, we now display ν_S in panel (c) and explicitly clarify in the caption the tuning of ν_S and the voltage bias used to emit quasiparticles or quasielectrons.

[Reviewer #3]

- I believe figure 1b should be more self-explanatory. Maybe writing 'e' or 'e/3' on the different. Figure is very hard to follow even for someone from the field and this is the first figure in the paper.

- Panel should be above/below so the reader can correlate immediately the transmission to noise.

[Reply]

We have split the previous Fig. 1b into the new Fig. 2a,b, with the previous inset showing ν_A now on top of the previous main panel with the same horizontal axis as requested by the reviewer.

We have simplified the figure by removing the unnecessary dashed lines corresponding to the asymptotic predictions of Eq. 1 (this was also done for Fig. 3a,b, Fig. 9a,b,c and Fig. 10a,b).

We also added graphic labels indicating the charge (and temperature) associated with the predictions displayed as continuous lines (the charge is now also graphically specified in Fig. 3c).

[Reviewer #3]

- The word bunching doesn't appear even once in the paper where it is actually a very common way in the community to think of this process – I believe the authors should address this point.

[Reply]

We have followed the reviewer's advice and now discuss the dual representation of the transmitted quasielectron as the bunching of three $e/3$ quasiparticles, as also graphically represented in the new Fig. 1a.

Other changes:

The manuscript was formatted according to the Nature Communications guideline. This includes a reworked abstract, new headlines, a displacement of the Methods section, a displacement and relabeling of the previous Extended Data Figures.

Detailed list of changes:

See marked manuscript. The new text is in red, except new headlines and new or updated references. The text removed is barred.

The main changes in the figures are:

- New Fig. 1a,b
- Previous Fig. 1a is replaced by Fig. 1c, with no inset and introducing graphically $\tau_{A,C}$.
- Previous Fig. 1b is replaced by Fig. 2a,b, with the previous inset shown as the separate panel (a) and with panel (b) from which the unnecessary asymptotic predictions are removed (dashed lines in previous Fig. 1b), now including an illustrative schematic of the device configuration and novel graphical labels for the displayed predictions.
- Fig. 3 (previous Fig. 2) now includes two illustrative schematics of the device configurations (panels (c),(d)), a graphical label indicating the charge associated with the different predictions (panel (c)), and unnecessary asymptotic predictions were removed (from panels (a),(b), dashed lines in same panels of previous version).
- Fig. 4 (previous Fig. 3) now includes separate graphical representations of the different device configurations compared in the main panel.
- Fig. 6 (previous Extended Data Fig. 2) now includes as an inset the previous separate table recapitulating the tank circuit parameters.
- Fig. 7 (previous Extended Data Fig. 3) now includes on display headlines mentioning the corresponding QPC.
- Fig. 9a,b,c and Fig 10a,b (previous Extended Data Fig. 5a,c,e and Extended Data Fig. 6a,b, respectively) do not include anymore the unnecessary asymptotic predictions (dashed lines in previous versions).

References: we added [26,31,32] and updated [27].

REVIEWER COMMENTS

Reviewer #1 (Remarks to the Author):

The authors have successfully answered all my questions and provided necessary changes in the revised manuscript. In particular, the new paragraph in the introductory part would be helpful for readers to understand the main results and share the novelty of this article. Thus, I recommend this article to be published in Nature communications.

Reviewer #2 (Remarks to the Author):

Authors have answered my doubts with some degree of clarity. I have very diligently looked at all the data, and somehow I noticed that the noise data deviates as there is change in slope for the transmission. I highly doubt that what the authors are claiming as Andreev scattering can very well be the result of such nonlinear qpc response. The fit in the graphs also deviates to some degree and their coincidence with the qpc response is very curious. On the basis of my current thoughts, I am afraid I am not being able to support the publication of this work in this current form.

Reviewer #3 (Remarks to the Author):

The authors have revised their manuscript extensively and made it more approachable and clearer for a reader which is outside of the immediate field.

Last comment I would like to add in my review is to ask the authors to demonstrate in a clearer way the differences from conventional Andreev reflection. This will also give the reader a bigger picture of similarities and differences from AR. The two clarifications I thought would be good to address (perhaps in the introduction) are:

- Spins. Spin conservation in this process. In AR the spin of the reflected hole is opposite from the spin of the impinging electron and the Cooper pair is in the singlet state. What are the different spins of the quasiparticles and how spin is conserved in this problem.
- Energies. In AR the impinging electron is impinging in an energy 'E' and the hole is reflected in energy '-E' so energy would be conserved in the process. Again, what is the scenario here?

Since the title and the paper state naturally the similarity to Andreev reflection I think it's imperative to show also the difference between these two processes. I believe the authors can show nicely the answer to these questions by splitting Fig1a and show the similarities and differences between AR and the process they are measuring in 2 analogues illustrations.

We would like to thank the reviewers for their suggestions and remarks helping us to further improve our manuscript.

Reviewer #1

The authors have successfully answered all my questions and provided necessary changes in the revised manuscript. In particular, the new paragraph in the introductory part would be helpful for readers to understand the main results and share the novelty of this article. Thus, I recommend this article to be published in Nature communications.

[Reply]

We thank the reviewer for his positive appreciation.

Reviewer #2

Authors have answered my doubts with some degree of clarity. I have very diligently looked at all the data, and somehow I noticed that the noise data deviates as there is change in slope for the transmission. I highly doubt that what the authors are claiming as Andreev scattering can very well be the result of such nonlinear qpc response. The fit in the graphs also deviates to some degree and their coincidence with the qpc response is very curious.

On the basis of my current thoughts, I am afraid I am not being able to support the publication of this work in this current form.

[Reply]

We thank the reviewer for his careful examination.

As the agreement is close to perfect in the main signal shown in Fig. 3a,c and also in the additional data shown in Fig. 9 in Methods (at an experimental accuracy that the reviewer previously found particularly high), we assume that the reviewer refers to the separate shot-noise characterization of the analyzer QPC shown in Fig. 2.

In the figure 2, we show that the analyzer QPC set in the tunneling regime favors the transfer of quasielectrons of charge e (which is then directly ascertained for the Andreev process itself from the shot noise data shown as blue symbols in Fig. 3c and Fig 9d,e,f).

The shot noise data in Fig. 2b follows the prediction of Eq. 1 for $e^*=e$, but only at sufficiently small voltage biases corresponding to low values of the transmission ratio τ_A (at $\tau_A < 0.3$ the deviations to the theory for $e^*=e$ remain small, comparable with the HEMTs calibration uncertainty visible in the figure through the difference between $\langle \delta I_T^2 \rangle$ and $\langle \delta I_R^2 \rangle$; note that in Fig. 3c $\tau_A < 0.2$). In contrast, a shot noise below the prediction for $e^*=e$ together with a slowdown of the growth of τ_A with the bias can be seen for larger τ_A ($> \sim 0.3$). We believe this is what the reviewer has in mind in his comment.

Such a behavior is however what theory predicts. Indeed, as the voltage bias is increased an initially small transmission ratio is predicted to grow (non-linearly, and slowing down at higher τ_A) and, simultaneously, the tunneling charge $e^*=e$ at low transmission is predicted to evolve toward a smaller tunneling charge $e^*=e/3$ (and thus less shot noise) at high transmission (see eg [10,11]).

This is pointed out in the manuscript, in the left column of page 3, below Eq. 1.

In response to the reviewer, we have also added this information in the caption of Fig. 2.

Reviewer #3

The authors have revised their manuscript extensively and made it more approachable and clearer for a reader which is outside of the immediate field.

Last comment I would like to add in my review is to ask the authors to demonstrate in a clearer way the differences from conventional Andreev reflection. This will also give the reader a bigger picture of similarities and differences from AR. The two clarifications I thought would be good to address (perhaps in the introduction) are:

- Spins. Spin conservation in this process. In AR the spin of the reflected hole is opposite from the spin of the impinging electron and the Cooper pair is in the singlet state. What are the different spins of the quasiparticles and how spin is conserved in this problem.

- Energies. In AR the impinging electron is impinging in an energy 'E' and the hole is reflected in energy '-E' so energy would be conserved in the process. Again, what is the scenario here?

Since the title and the paper state naturally the similarity to Andreev reflection I think it's imperative to show also the difference between these two processes. I believe the authors can show nicely the answer to these questions by splitting Fig1a and show the similarities and differences between AR and the process they are measuring in 2 analogues illustrations.

[Reply]

We thank the reviewer for his positive appreciation and his suggestions.

Following the reviewer's remarks, we have further extended our discussion on the relationship between the present Andreev-like process and the original Andreev mechanism taking place at a normal-superconductor interface. As the standard Andreev mechanism is not the focus of the present work, we believe it might be confusing and possibly disproportionate to show a specific schematic in Fig. 1. However, we have a paragraph in the introduction (second to last paragraph of page 1) dedicated to the comparison between the two mechanisms.

In response to the first reviewer's point on spin, we now point out in the last sentence of the dedicated paragraph that the present Andreev-like mechanism takes place in a fully spin-polarized electronic fluid. The overall conservation of charge, spin and density are thus directly connected. This differs with the standard Andreev process where two electrons of opposite spins are combined to form of a spin singlet Cooper pair, as now directly said.

Regarding the second point, the implication of energy conservation is also discussed in the dedicated paragraph following a suggestion of the reviewer. Furthermore, we have added a sentence to explicitly indicate that the reflected quasihole excitation has an energy lower than that of the incident quasiparticle. (Note that we speak of (quasi)hole excitations and thus positive energies.)

Other change: added reference [13].

REVIEWERS' COMMENTS

Reviewer #2 (Remarks to the Author):

The current version is satisfactory to some extent, hence I agree to publish the results with their explanations suitable for publishing in Nature communications.

Reviewer #3 (Remarks to the Author):

The authors have answered all my questions and comments. I recommend publication in the journal.